# A comparison of the effectiveness of functional MRI analysis methods for pain research: The new normal

Patrick W. Stroman [1,2,3]*, Howard J. M. Warren [1], Gabriela Ioachim [1], Jocelyn M. Powers [1], Kaitlin McNeil [1,4]

1 Centre for Neuroscience Studies, Queen's University, Kingston, Ontario, Canada, 2 Department of Biomedical and Molecular Sciences, Queen's University, Kingston, Ontario, Canada, 3 Department of Physics, Queen's University, Kingston, Ontario, Canada, 4 Royal Military College of Canada, Kingston, Ontario, Canada

☯ These authors contributed equally to this work.
* stromanp@queensu.ca

**Data Availability Statement:** All relevant data are within the manuscript and the public repository Figshare. Image data and associated study data can be found at the following URL and DOI. URL:

## Abstract

Studies of the neural basis of human pain processing present many challenges because of the subjective and variable nature of pain, and the inaccessibility of the central nervous system. Neuroimaging methods, such as functional magnetic resonance imaging (fMRI), have provided the ability to investigate these neural processes, and yet commonly used analysis methods may not be optimally adapted for studies of pain. Here we present a comparison of model-driven and data-driven analysis methods, specifically for the study of human pain processing. Methods are tested using data from healthy control participants in two previous studies, with separate data sets spanning the brain, and the brainstem and spinal cord. Data are analyzed by fitting time-series responses to predicted BOLD responses in order to identify significantly responding regions (model-driven), as well as with connectivity analyses (data-driven) based on temporal correlations between responses in spatially separated regions, and with connectivity analyses based on structural equation modeling, allowing for multiple source regions to explain the signal variations in each target region. The results are assessed in terms of the amount of signal variance that can be explained in each region, and in terms of the regions and connections that are identified as having BOLD responses of interest. The characteristics of BOLD responses in identified regions are also investigated. The results demonstrate that data-driven approaches are more effective than model-driven approaches for fMRI studies of pain.

## Introduction

Pain is complex and highly subjective, being an emotional and cognitive response to nociceptive signaling, and varying with a person's environment and mental state [1, 2]. Pain serves an important purpose of motivating us to protect ourselves from injury, and to protect an injury while it heals. However, it can have devastating effects when it is maladaptive and occurs in the

https://doi.org/10.6084/m9.figshare.13176860.v1.
DOI: 10.6084/m9.figshare.13176860.

**Funding:** Funding was provided by the Natural Sciences and Engineering Research Council of Canada (NSERC), grant number RGPIN/06221-2015. The funders had no role in study design, data collection and analysis, decision to publish, or preparation of the manuscript.

**Competing interests:** The authors have declared that no competing interests exist.

absence of noxious input, or is disproportionate to noxious or sensory stimuli. As a result, pain presents challenges for clinical and basic research. In spite of its impact, our understanding of the neural processes underlying pain is still quite limited [3].

In the 1960s major advances were realized as a result of electrophysiological studies in the brainstem (BS) and spinal cord (SC) [1, 4, 5]. These studies identified the importance of descending regulation of nociceptive responses in the SC, and their influence on the pain a person experiences. Pain research has been further advanced by the development of functional neuroimaging, such as functional magnetic resonance imaging (fMRI), but the focus of pain research with fMRI shifted to the cortex due to the technical challenges of imaging the BS and SC. While these studies have added substantially to our understanding of pain (for example [6–9]), it has become clear that cortical areas identified as being involved with pain processing are involved with many functions, such as salience, and are not specific to pain [8, 10]. Functional MRI methods have been adapted for the BS and SC and the studies to date have identified regions that are involved with pain processes. These studies have added to our understanding of descending regulation of the SC, and have shown that it is a dynamic process that is influenced by cognitive and emotional factors (placebo, nocebo, mood, music), and that blood oxygenation-level dependent (BOLD) signal variations can be detected in a resting-state, and in advance of noxious stimuli [11–33]. The latter were observed to be correlated with individual pain ratings to the stimuli which followed, and suggest the influence of pain anticipation in these regions [30, 31, 33, 34].

These studies also demonstrate the challenge for functional MRI analysis; how can we identify regions with BOLD responses to noxious stimuli, when we cannot predict the resulting BOLD responses in each person, and each situation? It is the differences in BOLD responses between individuals, or study groups, that are of interest for pain research, and these differences are not limited to the magnitude of the BOLD response with a fixed temporal pattern. The differences can include dynamic variations related to the myriad effects described above. It is possible that the limitations of using conventional model-based fMRI analysis methods (i.e. fitting time-series data to a predicted response) has resulted in important aspects of human pain responses going undetected. We therefore proposed to adapt data-driven fMRI analysis methods for pain studies, in order to detect the temporal patterns of BOLD responses. The objective of this study was to compare model-driven and data-driven fMRI analysis methods, using fMRI data from healthy participants in order to investigate the use of such methods, and identify their strengths and limitations. The data sets used span both the brain, and the BS and SC, in separate data sets, acquired with methods that were optimized for each region, and duplicate data sets are used in order to verify the performance of the methods that are compared.

## Materials and methods

### Functional MRI analysis approaches

Although the fMRI analysis methods to be compared are well-established and have been described in detail previously, we briefly describe the methods here in order to provide background for non-experts in fMRI theory, and to explain the approach and rationale for the present study.

**Model-driven analysis (fit to a predicted model).** One of the first fMRI analysis methods [35, 36], and still likely the most widely-used for task-based fMRI studies, consists of predicting BOLD response patterns of interest, and identifying which voxels have time-series responses that *sufficiently* match this predicted pattern. The concept is that we impose a model of the BOLD pattern we are looking for, and determine the anatomical locations where it occurs. In

the simplest model, a single predicted BOLD response, $P_{BOLD}$ (which is a function of time), can be modelled as the expected variation in signal over time, with an average value of zero, and a peak-to-peak amplitude of 1 (for simplicity). The measured time-series MRI signal from a voxel, S, (also a function of time) can then be modelled as $S = \beta_1 P_{BOLD} + \beta_0 + err$. Expressed in this form, $\beta_1$ represents the magnitude of the fit BOLD response, and $\beta_0$ is the average value of S. The "err" term is the residual signal variation that is not accounted for by the model. Calculating the fit of the measured data to the model determines the values of $\beta_1$ and $\beta_0$, and the standard errors of these values. The larger the values in the "err" term, the greater the standard errors of the β values (i.e. the worse the fit). The "null hypothesis" is that $\beta_1 = 0$, and if the probability of this being true is low enough, then the alternative hypothesis is inferred to be true, and S is concluded to have a significant component that corresponds with the predicted BOLD response. That is, the voxel response is significant, and $\beta_1$ is the magnitude of the predicted BOLD response. It is not necessary for the measured data to describe every detail of the predicted BOLD response pattern, but it is necessary to measure data at times with different predicted values (such as high and low values).

**Data-driven analysis (connectivity analysis).** The alternative to modeling the BOLD response and identifying regions where it occurs, is to identify MRI signal variations in time which are related across spatially separated anatomical regions, and are therefore inferred to be physiologically relevant. In the context of fMRI data, these signal variations are assumed to be BOLD if they occur consistently, and in relevant anatomical regions. Even though the BOLD pattern(s) may be unknown or unpredictable, they can be identified based on the relationships between regions and anatomical information. While these methods are most often used to analyze resting-state fMRI data, they can also be applied to task-based data. Relationships between pain processing and BOLD responses or dynamic variations in connectivity can be identified with retrospective analysis of how they relate to the timing of noxious stimuli in a task-based study. While there are a number of data-driven analysis methods available [37], for the present comparison we focus on functional connectivity analyses. Connectivity analyses are based on the concept that BOLD signal variations are related to changes in metabolic demand, which is driven by pre-synaptic input signaling to a region [38]. If two regions have temporally related input signaling, then their BOLD responses are similarly related.

The simplest form of connectivity analysis is to calculate the temporal correlation between all pairs of voxels in an fMRI data set [39, 40]. This produces a large number of results, which must be assessed in relation to the anatomical locations of significantly connected voxels. This approach can be simplified by first identifying clusters of voxels which are spatially contiguous and have similar MRI time-series responses (and therefore similar function). Correlations can then be computed between all pairs of clusters. Similarly, the analysis can be limited to anatomical regions of interest, and can be applied on a voxel-by-voxel or cluster-by-cluster level.

However, this method imposes the assumption that a significant portion of the variance in a voxel/cluster time-series response can be explained by the time-series response in a single other region. In order to allow more flexibility, the time-series response in a region may be explained by the responses in two, or more, other regions. For example, if region A has a time-series response, $S_A$, and we want to test if this response can be explained by two other regions, B and C, then we can use the model $S_A = \beta_{BA} S_B + \beta_{CA} S_C + err$. This method is a form of Structural Equation Modeling (SEM) [41, 42] and requires some *a priori* knowledge (described below) of the anatomical regions that we want to model as the "target" regions and the "source" regions. The resulting fit provides the values of $\beta_{BA}$ and $\beta_{CA}$, which reflect the connectivity to A, from B and C, respectively. Fitting BOLD responses in a target region to BOLD responses in multiple sources, guided by the known neuroanatomy, enables the direction of the connectivity to be inferred. For example, the value of $\beta_{BA}$ with A as the target and B as one

of the sources, is generally expected to be different than the value of $\beta_{AB}$, with B as the target and A as one of the sources. However, the calculated connection strengths are estimates, as they are influenced by whether or not the time-series responses $S_B$ and $S_C$ are independent, or are themselves correlated. The added flexibility of the method also adds to the possible complexity of the results, with the potential for more information being obtained.

## Comparisons of analysis approaches

In the present study, model-driven and data-driven analysis methods were compared for the purpose of detecting pain processing in fMRI data from healthy human participants, in data sets from two prior studies which each spanned the brain and the BS/SC. However, the comparisons were limited to a selected set of analysis parameters for practical purposes. It is not possible to test every permutation of all methods. Functional MRI data from healthy participants were pre-processed to remove noise and confounds as much as possible. Clusters of voxels were identified within selected regions of interest based on the entire data set. Once the clusters were defined, cluster data were extracted for each participant/condition. The fMRI data acquisition, pre-processing, and clustering methods, are detailed below. All of the analysis methods being compared were applied to the same cluster data. The methods include 1) Model-driven fit to a predicted BOLD response, 2) Data-driven correlation between clusters, 3) Data-driven SEM with one source region, and 4) Data-driven SEM with two source regions. The significance of results was assessed in terms of the whether or not group-level fit parameters were significantly different than zero, whether connectivity accounted for a significant amount of variance in each region, and whether fit parameters were significantly correlated with pain ratings. The methods were compared using the amount of signal variance that could be explained in each region, and the numbers of regions that were identified as having BOLD responses.

## Data sets from prior studies

Data for this comparison of analysis methods were anonymized data obtained from the healthy control groups in two previous fMRI studies by our lab. These studies were reviewed and approved by the Health Sciences Research Ethics board at Queen's University. All participants provided informed written consent, prior to participating.

The data from each participant, in both studies, were obtained spanning the entire brain in one imaging session, and spanning the BS and cervical SC in another session. Some participants either did not return for two sessions or technical difficulties resulted in data being collected from only one region (brain or BS/SC). Although these studies shared common features with respect to their aims and study designs, there are some differences as detailed below. The datasets were not combined, but rather were used as duplicate independent analyses to confirm our findings. In order to differentiate the two datasets, the first study, which was conducted between 2013–2014, will be referred to as 'Study 1', and the second study, conducted between 2018–2019, will be referred to as 'Study 2'. Full details of the methods used for Study 1 have been previously published [11].

## Participant recruitment

**Study 1.** Healthy women were recruited from the local community and completed the study as a control group. All participants were free of previous neurological injury or disease, and were free of any contraindications for MRI. The study group consisted of 15 women (age range = 21–55, average 39.1 ± 10.2 years (mean ± std)).

**Study 2.** Healthy women were again recruited from the local community, and completed the study as a control group, and were again free of any previous neurological injury or disease, or any contraindications for MRI. This group consists of 18 women (age range = 21–59, average 36 ± 11.3 years).

## Participant characteristics and study preparation

Participants in both studies were characterized by completing questionnaires to assess anxiety, depression, pain catastrophizing, social desirability, and health-related factors, but these data are not used for the present analysis. Prior to fMRI data collection, each participant underwent a 1-hour training session, during which they were introduced to the experimental pain stimulus and study design, and were trained how to rate their pain using a standardized numerical pain intensity scale (NPS). The scale ranges from 0 to 100 in increments of 5, with verbal descriptors at increments of 10 [11, 43]. In both studies, the stimulus consisted of heat applied briefly to the skin overlying the thenar eminence (base of the thumb) on the right hand. The stimulus devices, temperatures, and timings, were different for the two studies, as detailed below. A series of calibration tests with varying stimulus temperatures (between 40˚C and 52˚C) were conducted so that participants could become accustomed to the thermal stimulation, and study procedures. The stimulus temperature was calibrated in order to elicit a rating of 50 ± 10 NPS units for the participant. The goal was to have every participant subjectively feel a similar intensity of pain. Participants also underwent a practice fMRI run in a mockup of our MRI system in order to add to their familiarity with the procedures and environment, and reduce potential anxiety. The pain ratings reported by participants consistently varied between the training and imaging portions of the study, and varied across fMRI acquisitions, resulting in a range of pain ratings in spite of the training and calibration efforts.

## FMRI paradigm

**Study 1.** Heat stimuli were applied to the hand by means of an MRI-compatible Peltier thermode (Medoc®, Ramat Yishai, Israel), which was attached to the participant's right hand, overlying the thenar eminence. During heat stimulation, the temperature was rapidly increased and decreased under computer control. Imaging runs were separated into two study conditions: temporal summation of second pain (TSSP) and TSSP-Control (TSSP-C). Only the data from the TSSP condition are used for the present comparison of methods. Repeated imaging runs for each condition were implemented in a randomized order, and a minimum of 2 minutes of rest was given between each run. For the TSSP condition, 11 heat spikes with the previously calibrated temperature to elicit a rating of 50 ± 10 NPS units were applied every 3 seconds. The TSSP condition's stimulation period was preceded by a 52 second rest period and followed by a 65 second rest period (Fig 1). Similar to the training session, participants viewed instructions on a rear-projection screen which notified them when a new scan was about to begin, when the application of the heat stimulus would begin, and when to report their ratings for the first and last heat contacts (Fig 1).

**Study 2.** Heat stimuli were applied to the participant's right hand overlying the thenar eminence, by means of an MRI-compatible Robotic Contact-Heat Thermal Heat Stimulator (RTS-2; SpinalMap Inc., Kingston, Ontario). This device pneumatically raises and lowers a heated aluminum thermode to make contact with the participants' skin, and is precisely controlled using software written in MATLAB (Mathworks Inc., Natick, MA). A 'threat/safety' paradigm was employed, in which participants were unaware at the beginning of each run whether or not a painful stimulus would be applied. The imaging runs were separated into two study conditions: 'Stimulation' and 'No-Stimulation'. Repeated imaging runs for each

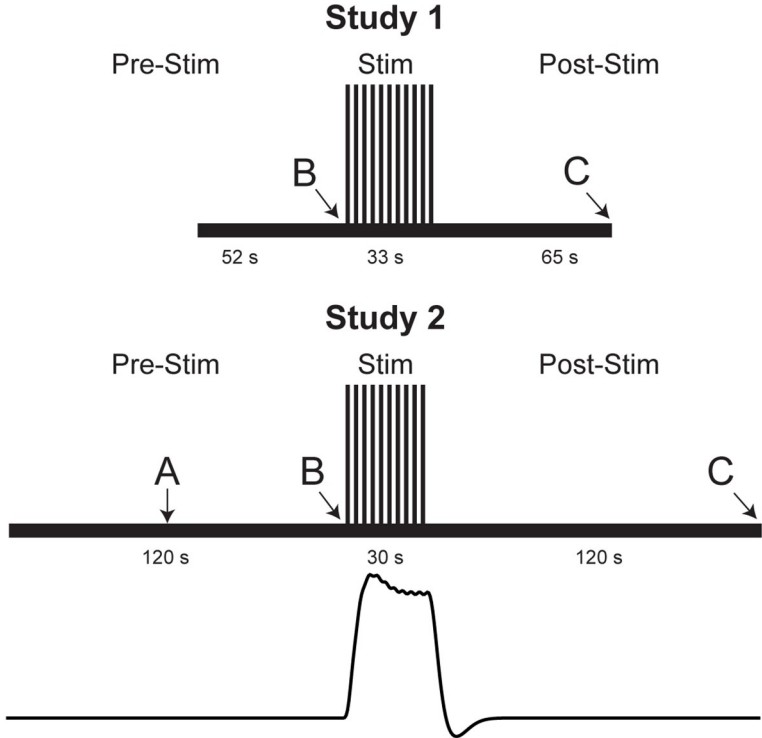

Predicted BOLD response based on the stimulation paradigms

**Fig 1. FMRI paradigms for Study 1 and Study 2.** Functional imaging runs in Study 1 were 150 seconds long, and the heat pain was administered as 11 consecutive heat spikes over a span of 33 seconds. Functional imaging runs in Study 2 were 270 seconds long, and the heat pain was administered as 10 brief heat contacts over a span of 30 seconds. Letters indicate different prompts being delivered to participants over the rear-projection screen. **A** = "you will feel heat." **B** = "the pain stimulus will begin in 3…2…1…" **C** = "please provide your first and last pain ratings." The predicted BOLD response based on the stimulation paradigm is also shown. However, the data sets from each participant consist of between 4 and 6 repeated runs with this stimulation paradigm.

condition were obtained in a randomized order, and a minimum of 2 minutes was given between stimulation periods in successive runs. A rear-projection screen was used to inform the participants when the next imaging run would begin (Fig 1). After 60 seconds of baseline scanning, participants were informed whether that run would involve the thermal heat stimulation (Stimulation) or not (No-Stimulation). If it was a Stimulation run, the participants were informed when the stimulation was to begin at the 120 second mark, and then 10 heat contacts at the calibrated temperature were administered over 30 seconds. This 30 second stimulation period was followed by a 120 second rest period, for a total time of 4 minutes and 30 seconds (Fig 1). At the end of the Stimulation runs, the participants were prompted to provide their pain ratings for the first and last heat contacts, via an intercom. Only the data from the Stimulation condition was used for the analyses in the present study.

## FMRI data acquisition

All image data were acquired using a 3 tesla whole-body MRI system (Siemens Magnetom Trio; Siemens, Erlangen, Germany). For all studies, participants were positioned supine and were supported by foam padding as needed to ensure comfort and minimize bulk body movement. Imaging methods were optimized for each region (brain, or BS/SC), due to the different imaging challenges, and were acquired with $T_2^*$-weighted imaging in the brain, and $T_2$-

weighted imaging in the BS and SC, in order to provide an optimal balance of image quality and BOLD sensitivity in both regions [31, 44, 45]. Although $T_2$-weighted imaging for BOLD sensitivity was demonstrated in some of the earliest fMRI studies [46, 47] it is rarely used for brain fMRI, but it is well-established for BS/SC fMRI [44]. Initial localizer images were acquired in three planes as a reference for slice positioning for subsequent fMRI studies.

**Study 1 brain fMRI.** Functional images were acquired in 49 contiguous axial slices oriented parallel to the anterior commissure-posterior commissure (AC-PC) line using a $T_2^*$-weighted gradient-echo echo-planar imaging (GE-EPI) sequence (TR = 3 s, TE = 30 ms, Flip Angle = 90˚, FOV = 192 mm x 192 mm, Matrix = 64 x 64, Resolution = 3 x 3 x 3 $mm^3$). A 12-channel head coil was used for detection of the MRI signal, with a body coil for transmission of RF pulses. A total of 50 volumes were acquired for each imaging run. Five runs of the same type were combined for each fMRI data set.

**Study 2 brain fMRI.** Functional images were acquired in 66 contiguous axial slices using a $T_2^*$-weighted GE-EPI sequence (TR = 2 s, TE = 30 ms, Flip Angle = 84˚, Multiband = 3, 7/8 Partial Fourier, FOV = 180 mm x 180 mm, Matrix = 90 x 90, Resolution = 2 x 2 x 2 $mm^3$). A 32-channel head coil was used for detection of the MR signal, with a body coil for transmission of RF pulses. A total of 135 volumes were acquired for each imaging run. Three to five runs of the same type were combined for each fMRI data set.

**Study 1 and Study 2, brainstem and spinal cord fMRI.** Functional MRI data were acquired with a $T_2$-weighted half-fourier single-shot fast spin-echo (HASTE) sequence. Data were acquired in 9 contiguous sagittal slices with a repetition time (TR) of 0.75 sec/slice, an echo time of 76 msec to optimize the $T_2$-weighted BOLD sensitivity, and a $28 \times 21$ cm field-of-view with $1.5 \times 1.5 \times 2$ $mm^3$ resolution [45]. The imaging volume spanned from the T1 vertebra to above the thalamus, and spanned the entire cervical SC and BS left-to-right. Data were acquired using the upper elements of a spine receiver-array coil, a posterior neck coil, and the posterior half of a 12-channel head coil. The receiver elements were adjusted based on the participant's size, as needed. A body coil was used for transmitting radio-frequency (RF) excitation pulses. In Study 1, a total of 138 volumes were acquired for each condition (over 6 repeated runs). In Study 2, a total of 200 volumes were acquired for each condition (over 5 repeated runs). The image quality was enhanced by means of spatial suppression pulses anterior to the spine to reduce motion artefacts caused by breathing, swallowing, etc, and motion compensating gradients in the head-foot direction.

## FMRI data preprocessing

**Brain fMRI data.** Data from both studies were preprocessed using SPM12 software (Wellcome Institute of Cognitive Neurology, London, UK). Data from both studies were preprocessed using SPM12 software (Wellcome Institute of Cognitive Neurology, London, UK). The data were converted from DICOM to NIfTI format, and were then co-registered to the third volume in the time-series to correct for motion, and motion parameters were retained for later noise modeling. We considered motion exceeding one voxel width or 5 degrees of rotation to be excessive, and no data sets were discarded because of excess motion. Slice-timing correction was applied by interpolating to the middle of each volume acquisition period. Spatial normalization was then applied to the data from each participant by first normalizing the fMRI data to the anatomical images from the same participant, to make use of the better data quality of the anatomical images, and then normalizing the anatomical images to the MNI template. The complete transformation needed to map the fMRI data to the MNI template (Montreal Neurological Institute, Montreal, Quebec) was thus determined. The first two volumes of each run were discarded to avoid periods without consistent $T_1$-weighting, and the remaining time-

series responses for each voxel were converted to a percent signal change from the time-series average.

**BS/SC fMRI data.** Functional images were preprocessed using our freely-shared software "SpinalFMRI9", written in MATLAB (The Mathworks Inc., Natick, MA) that has been used extensively in prior SC and BS fMRI studies [11, 48–50]. Pre-processing steps included conversion to NIfTI format, co-registration using the Medical Image Registration Toolbox (MIRT) [51], interpolation to 1 mm³ resolution, and spatial normalization to a pre-defined anatomical template based on 300 healthy participants, as described below, using normalization methods described previously [30, 33, 34, 44, 52]. Physiological noise was modeled, and then removed from the data, based on peripheral pulse recordings, bulk motion, and global signal variations in white matter. Our methods of removing physiological noise are highly effective and have been validated by quantifying the contribution from each source of noise, and making comparisons with data from cadavers [31, 45]. Finally, the first two volumes of each run were discarded to avoid periods without consistent $T_1$-weighting, and the remaining time-series responses for each voxel were converted to a percent signal change from the time-series average.

## Anatomical templates and region maps

We have combined region maps and anatomical reference images (templates) across the brain and SC regions (the entire CNS) as described by De Leener et al. [53]. An anatomical reference template was thus created by combining the PAM50 template [53] of the SC, and the MNI152 template of the BS/brain. The resulting map was interpolated to 0.5 mm cubic voxels, and can be scaled to lower resolution as needed. For our purposes, an anatomical reference image of the cervical SC and BS region (spanning where the two existing templates join) was also created. This was done by spatially normalizing fMRI data from 1440 fMRI data sets in 300 healthy participants to the combined template, and averaging the normalized images to create a single template.

Corresponding anatomical region-of-interest maps (0.5 mm isotropic resolution) were also defined across the entire CNS by combining probabilistic regions maps from multiple sources, and region maps based on anatomical atlases and published descriptions. Brain regions were identified primarily with the region maps provided in the CONN15e software package [1, 54, 55]. SC gray matter and white matter maps were obtained from PAM50 template in "The Spinal Cord Toolbox" [56]. In addition, the SC was divided into segments, with positions based on the anatomical study done by Lang and Bartram [57], and segments were divided into right/left and anterior/posterior quadrants. BS regions not included in the CONN15e region map were added based on examples and anatomical descriptions [1, 58–62], freely shared atlases as described by Pauli et al. [63] (https://identifiers.org/neurovault.collection:3145), Keren et al. [64], and Harvard atlases [65]. The resulting region maps are thus probabilistic, and indicate the likely location of anatomical regions of interest within each data set, after it has been spatially normalized.

## Definition of clusters

The anatomical regions to be used for the SEM analysis were identified in spatially normalized data using the pre-defined anatomical reference described above. As it is not expected that entire anatomical regions are uniformly involved in pain responses, regions were first divided into 7 clusters based on the fMRI data. That is, the clustering was based on BOLD responses as opposed to assumed anatomical sub-divisions. Voxels within each region were first identified, and clustering was carried out using a k-means method. This was applied to data from each

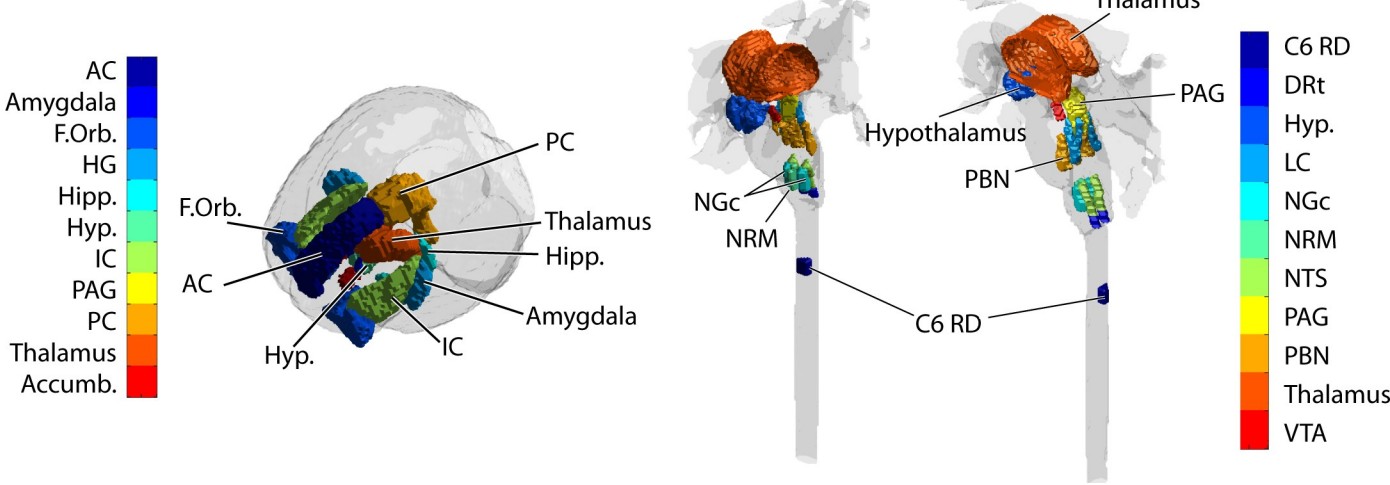

**Fig 2.** Anatomical regions included in brain (left), and BS and SC (right), data analysis.

study separately, which was concatenated across all runs in all participants. The cluster definitions are thus based on similarity of the measured time-series responses across the entire study group. This procedure was repeated for each region of interest, and the same cluster definitions were used for analyzing data from Study 1 and Study 2, but different clusters were used for brain data, and BS/SC data, because of the different imaging parameters and anatomical regions. The choice of 7 clusters for each region was based on previous studies which showed that this number provided a suitable balance of resulting cluster sizes and flexibility in the separation of anatomical sub-regions. The regions used are displayed in Fig 2 by generating a 3D plot in MATLAB, rendering the region surfaces using the MATLAB functions "isosurface", "patch", and "isonormals", and assigning each region a different colour. The 2D figures were then captured in PDF format from the 3D plots and paste into Adobe Illustrator for adding text annotations and colour bars. For brain fMRI data the regions include the anterior cingulate (AC), amygdala, frontal orbital region (FOrb), heschel's gyrus (HG), the hippocampus (Hipp), hypothalamus (Hyp), insular cortex (IC), periaqueductal gray region (PAG), posterior cingulate (PC), thalamus, and nucleus accumbens (Accumb.). For BS/SC fMRI data the regions include the right dorsal region of the 6[th] cervical SC segment (C6RD), the dorsal reticular nucleus of the medulla (DRt), the hypothalamus, locus coeruleus (LC), nucleus gigantocellularis (NGc), nucleus raphe magnus (NRM), nucleus tractus solitarius (NTS), the PAG, parabrachial nuclei (PBN, medial and lateral divisions), thalamus, and ventral tegmental area (VTA).

## Application of the data analysis methods and comparison of results

Following pre-processing steps described above, clusters were defined for selected regions of interest which are known/suspected of being involved with aspects of pain. Voxel data were then extracted and average time-series responses were computed for each cluster, in each run, from each participant. The following analyses were then applied:

1. Data were fit to predicted BOLD response (Fig 1) using a general linear model (GLM), and the significance of β-values were determined (i.e. must be significantly different than zero, which requires that the fit accounts for a significant amount of variance). The predicted response was determined by convolving the canonical hemodynamic response function

with the timing of the stimulation paradigm [66]. Analyses were applied to data from each participant concatenated into one large data set, to data from all participants averaged, and also to data from each participant individually, with repeated runs concatenated. The number of time points used for each calculation are listed in Table 1.

2. Connectivity was computed based on the correlation between all possible pairs of clusters from different regions, and significance was based on Z-values (the correlation must account for a significant amount of the variance). Data from all participants were concatenated for selected time periods, and were also analyzed for each participant individually, with repeated runs concatenated. Dynamic connectivity was calculated using a sliding window spanning 30 second epochs for brain data (10 or 15 volumes, Study1 and Study 2 respectively), and 45 second epochs for BS/SC data, or 7 volumes. The number of time points used for each calculation are listed in Table 1.

3. Connectivity was computed using SEM with 1 or 2 source regions in separate analyses, with significance based on the amount of variance explained in the target region (expressed as Z values), and on the significance of β-values. Data from all participants were concatenated for selected time periods, and were also analyzed for each participant individually, with repeated runs concatenated. Significance was inferred if the fit explained a significant amount of variance in the target region, and all connectivity weighting factors (β-values) were significantly different than zero, and also the F-test for each connection was significant (all connections contributed significantly to the fit). Again, the number of time points used for each calculation, for each target or source region, are listed in Table 1. The SEM models that were used, are shown in S1 Fig.

4. β-values determined from GLM fits, and SEM fits, were tested for correlation with pain ratings in each participant (connectivity correlation, R, values were not included because these values are not linearly related to strength of the connection).

5. The performance of analysis methods was compared based on the average amount of variance explained in each cluster.

6. Results were assessed by investigating the characteristics of the BOLD responses in regions identified by the previous analyses, and comparing the temporal pattern of BOLD responses across regions, data sets, and to the stimulation paradigms.

## Statistical thresholds

Statistical thresholds used to infer significance were corrected for multiple comparisons, in every case. In order to confirm that suitable thresholds were used, all analysis methods were applied to replicated data sets consisting of random values, to create "null" data sets in which

**Table 1. The number of imaging volumes (i.e. time points) used for each calculation with each analysis method, at the group and individual levels.**

| | GLM fit to Model | | Dynamic Connectivity (both correlation and SEM methods) | |
|---|---|---|---|---|
| | **Group** | **Individual (average/person)** | **Group** | **Individual (average/person)** |
| **Study 1 Brain** | 3600 | 240 | 750 | 50 |
| **Study 2 Brain** | 8316 | 462 | 945 | 53 |
| **Study 1 BS/SC** | 1848 | 123 | 616 | 41 |
| **Study 2 BS/SC** | 2622 | 146 | 483 | 27 |

Calculations were repeated at multiple time points for dynamic analyses. The same number of volumes were used for each target and source region for SEM analyses.

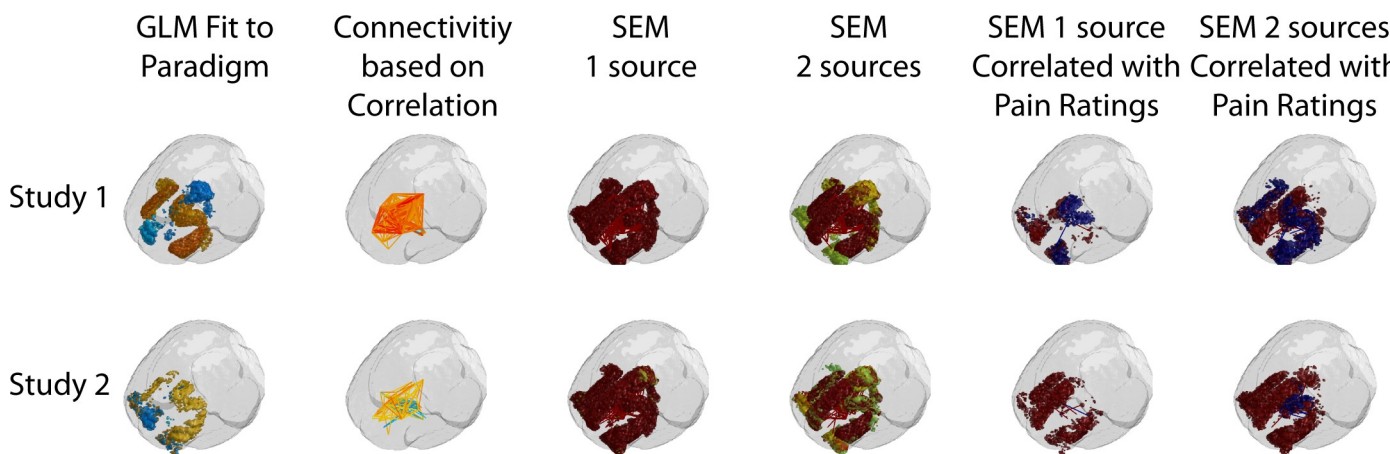

**Fig 3. Anatomical regions identified with the analysis methods tested in brain regions.** Regions are shown for the GLM fit, connections are shown for correlation results, and both regions and connections are shown for SEM results.

no significant values should be detected. Statistical distributions were thus determined, and statistical thresholds were identified to ensure that there was less than a 5% chance of a single false-positive (Type I) result occurring in any analysis (i.e. family-wise-error corrected $p < 0.05$).

## Results

Each analysis method identified significant BOLD responses in regions, or as connectivity between regions, across the two sets of study data, in both brain and BS/SC regions. The results of the analyses are summarized in Figs 3 and 4. These figures were generated using the methods described above for Fig 2, with the addition of significant connections between regions indicated with lines plotted in MATLAB. A larger number of regions/clusters were identified

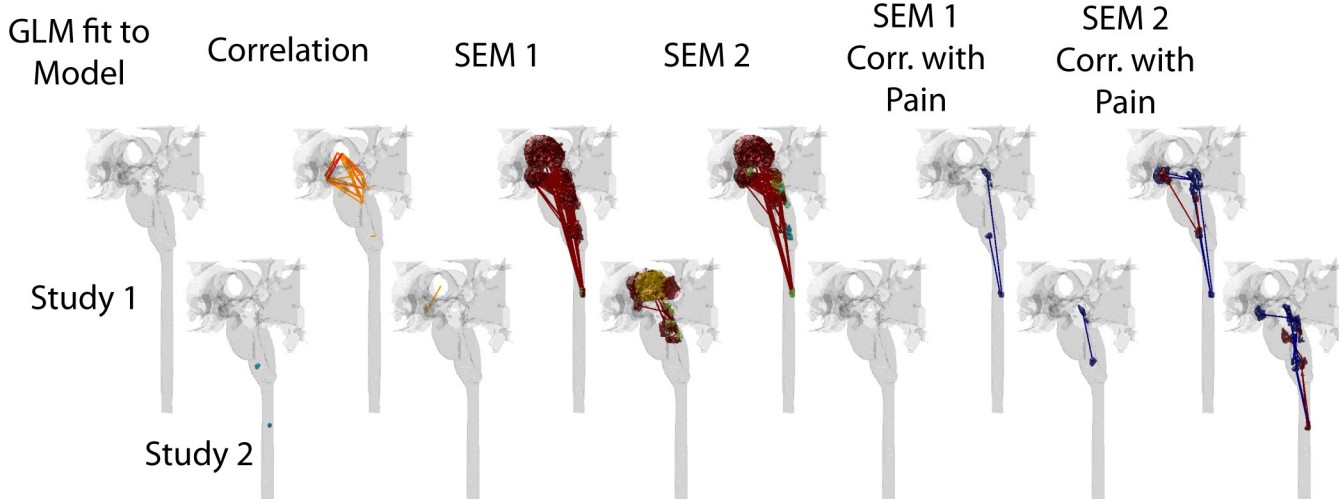

**Fig 4. Anatomical regions identified with the analysis methods tested in BS and SC regions.** Regions are shown for the GLM fit, connections are shown for correlation results, and both regions and connections are shown for SEM results. "GLM fit to Model" refers to the model-driven analysis, "Correlation" indicates connectivity analysis based on temporal correlation, SEM 1 and SEM 2 refer to structural equation modeling with 1 and 2 source regions, respectively, and "Corr. with Pain" indicates SEM results (β-values) which are significantly correlated with pain ratings from each participant.

**Table 2. Summary of the regions identified with each of the analysis methods.**

| | Brain fMRI | | BS/SC fMRI | |
| --- | --- | --- | --- | --- |
| | Study 1 | Study 2 | Study 1 | Study 2 |
| **GLM fit to predicted response** | AC, PC, IC, Thalamus, Accumbens, HG, Hypothalamus, PAG | AC, IC, Thalamus, Accumbens, FOrb | - | NRM, C6RD |
| **Connectivity based on Correlation** | All 11 regions | All 11 regions | Thalamus, Hypothalamus, PAG, NTS, PBN, LC, NGC | Thalamus, Hypothalamus |
| **Connectivity with SEM, 1 source** | All 11 regions | All 11 regions | Thalamus, Hypothalamus, PAG, NTS, PBN, LC, NRM, NGc, DRt, C6RD | Thalamus, Hypothalamus, PAG, PBN, LC |
| **Connectivity with SEM, 2 sources** | All 11 regions | All 11 regions | Thalamus, Hypothalamus, PAG, NTS, PBN, LC, NGc, C6RD | - |
| **Correlation between pain ratings and SEM connectivity values** | All 11 regions | 10 regions, only HG was not identified | Hypothalamus, PAG, LC, NRM, NGC, C6RD | Hypothalamus, PAG, NTS, PBN, LC, NRM, NGC, DRt, C6RD |

For each data type (brain, or BS/SC) 11 regions were included, and each region was separated into 7 clusters, as described in the Methods. The regions included for brain data were: AC, PC, IC, Thalamus, Accumbens, HG, Hypothalamus, PAG, FOrb, Hippocampus, and Amygdala. The regions included for BS/SC data were: Thalamus, Hypothalamus, VTA, PAG, NTS, PBN, LC, NRM, NGc, DRt, and C6RD. No regions were identified as having fit parameters (β-values) or connectivity values, that were correlated with pain ratings, and so they are not included in the table.

in all data sets by means of data-driven approaches, including correlations between time-series responses, and structural equation modeling (SEM), than with the model-driven GLM fit to a predicted response. A number of regions were also observed to have connectivity weighting values, measured with SEM, that are significantly correlated with individual pain ratings, also as shown in Figs 3 and 4. The regions identified with each method are summarized in Table 2, for brevity. Details of the regions and connections identified with each analysis method are listed in S1 through S10 Tables. The average amount of variance explained per cluster, with each analysis method, is plotted in Fig 5.

Time-series responses were extracted from selected regions, based on the analysis results, in order to examine their physiological relevance and relationships with the stimulation paradigms. Examples of responses across brain and BS regions in Study 1 are shown in Fig 6, for the anterior cingulate (AC), insula (IC), and nucleus raphe magnus (NRM). A larger set of time-series responses are shown in S2 Fig, from all 4 data sets, for selected regions including the AC, IC, hypothalamus, thalamus, PAG, NGc, and NRM. Examples of dynamic variations in connectivity detected with SEM are shown in Fig 7, for the selected connection of PAG→Hypothalamus. This connection was chosen because it occurs in all of the data sets that were analyzed. Clusters shown were selected from BS/SC data, and similar source and target clusters were selected for comparison with brain fMRI data. Because the cluster definitions are different for the two data sets, and the different acquisition parameters such as spatial resolution, the exact same clusters are not shown, but have centers that are within 11.2 mm for the sources, and within 8.7 mm for the targets.

## Discussion

BOLD responses were identified in regions across the brain, BS and SC, with each of the analysis methods. The regions identified by fitting the data to a predicted response (model-driven) were consistently also identified by the data-driven approaches, which showed additional regions as well. The conclusion that the regions identified have signal variations that are BOLD and are physiologically relevant, as opposed to being the result of bulk motion or

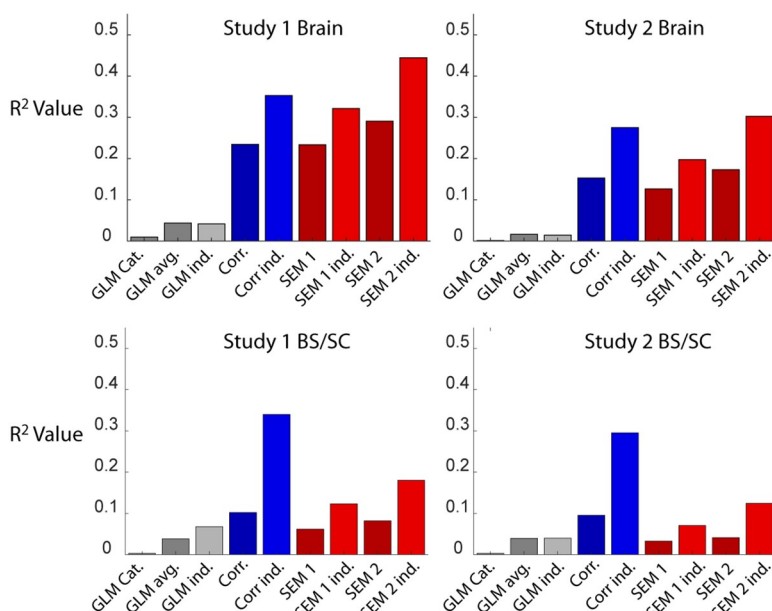

Average Variance Explained in Each Region/Cluster, using Selected Analysis Methods

**Fig 5. Plots of the average amount of variance explained in each region included in the analysis, for each of the analysis methods being compared.** Abbreviations: GLM; general linear model fit of the data to a predicted BOLD response, Corr.; connectivity calculated by means of temporal correlation, SEM1 and SEM2; structural equation modeling with 1 or 2 source regions, respectively. Variations are indicated as follows: "cat"; concatenated data across the group, "avg"; data averaged across the group, "ind."; analysis applied to data from each individual participant.

physiological noise, is supported by the fact that the responses are consistent across repeated fMRI acquisitions in each participant, within each group of 15–18 participants, and either match predicted responses or are significantly coordinated across spatially distinct regions. This conclusion is further supported by the consistency of results across independent data sets, and across different acquisition methods in regions of overlap between the brain and BS/SC studies (illustrated in Figs 3, 4 and 6). Dynamic variations in connectivity between regions were also observed to be relatively consistent across studies, such as the PAG→Hypothalamus connection in Fig 7. Although some degree of variation is to be expected with the different methods and study conditions that were employed, the conclusions that can be drawn from these results are quite robust, in terms of the regions involved, and how signaling is coordinated between regions.

The amount of signal variance that could be explained in each region varied across the analysis methods used for this comparison, but showed a consistent trend across the 4 sets of data (Fig 5). The amount of explained variance was consistently higher with data-driven methods, than with the model-driven fit to a predicted BOLD response. The greatest amount of variance was explained, in most cases, with connectivity based on correlation between regions. The exception was the SEM method with 2 source regions, which explained the most variance in brain regions. The ability to explain the signal variance, and thus identify BOLD time-series responses, demonstrates the effectiveness of each analysis approach for the selected fMRI data sets which involved noxious stimulation. The results consistently indicate that data-driven approaches are more effective than model-driven approaches, and further indicate that detecting connectivity based on temporal correlations may be more effective than SEM. However, the incorporation of anatomical information into the SEM method of connectivity analysis

# BOLD Response Details from Study 1

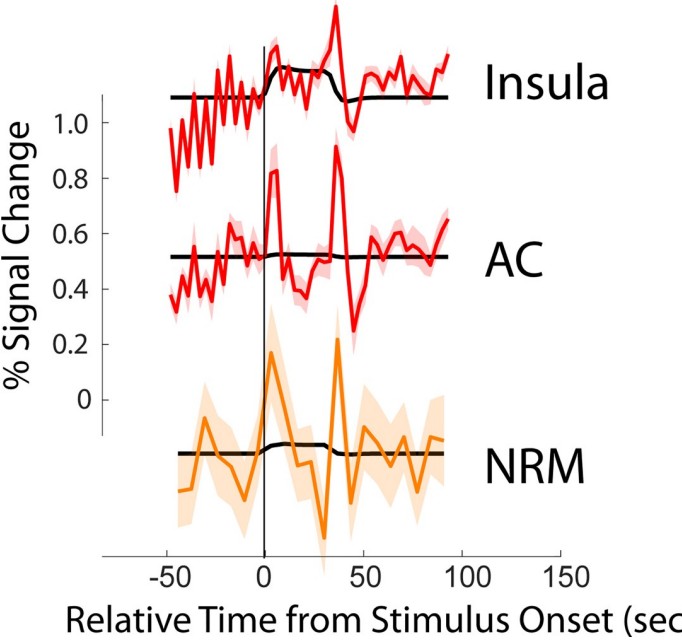

**Fig 6. Details of BOLD responses in selected regions from Study 1.** The three plots are offset vertically for clarity. The vertical axis indicates the relative scale for each of the 3 plots. The horizontal axis indicates time, with the origin at onset of stimulation. The vertical line indicates when participants were informed of which type of stimulation to expect in Study 2. Lines plotted in red are from brain fMRI data, and the line plotted in orange is from BS/SC data. The fit model paradigm to each time-series response is also shown in black on each plot.

# Dynamic PAG to Hypothalamus Connectivity detected with SEM

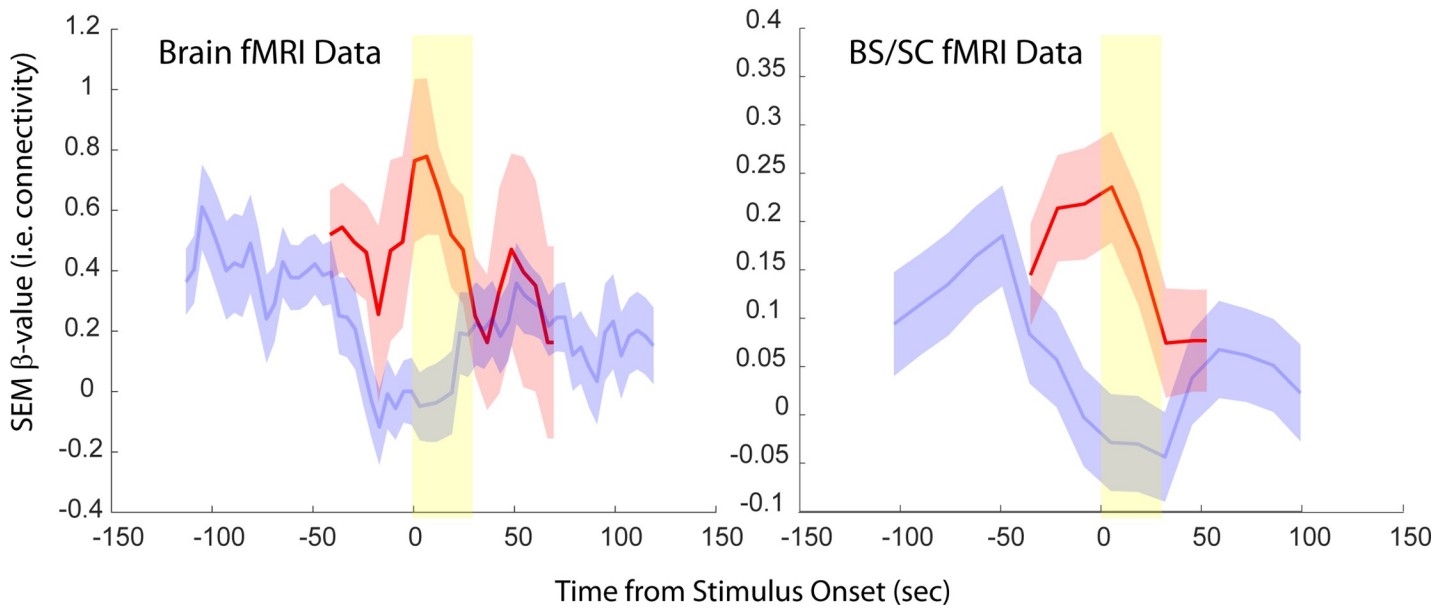

**Fig 7. Examples of dynamic variations in the PAG→Hypothalamus connection, as identified with SEM analysis with 2 sources.** Results are shown for data obtained in the brain, and in the BS/SC for Study 1 (red) and Study 2 (blue). The yellow band indicates the stimulation period. The error bars indicate the standard error of the β-values. Similar source and target clusters were selected for the two regions spanned by the data, and the results were within 11.2 mm for the sources, and with 8.7 mm for the targets.

provides the advantage of greater detail in the results that can be obtained. By modeling the possible sources of input to each region, the directionality of connectivity can be inferred. Whereas correlation does not show any information about the direction of influence, the linear fit between a source and a target will typically result in different weighting factors (i.e. connectivity strengths) if the assignment of source and target are reversed. Including more than two source regions could potentially explain even more of the variance in each target region. Conversely, the need to assume an anatomical model for SEM may be a disadvantage for studies with the aim of identifying regions involved, as opposed to investigating the characteristics of BOLD responses in predetermined regions-of-interest.

An important aspect of the data-driven methods for studies of pain responses is that they can demonstrate dynamic variations in coordinated signaling between regions. This allows more degrees of freedom, which contributed to the greater amount of variance that could be explained. Similarly, including multiple source regions to explain the signal variance in a target region, allows more degrees of freedom with the SEM method. This flexibility is important in the context of pain research, given that neural signaling between regions is expected to vary between periods of anticipating pain, experiencing pain, knowing the pain has passed, etc [32, 33]. Moreover, the known neuroanatomy of regions involved with pain processing is a complex network, and each region receives input from multiple sources (1). The flexibility of the data-driven approach thus enables more accurate modeling of the expected neurophysiology, and neuroanatomy, of pain networks.

Investigations of the BOLD response characteristics (Fig 6 for a summary, S2 Fig for more details) demonstrate why the predicted BOLD response, based on the peripheral stimulation paradigm, is not an accurate model. The typical modeled BOLD response would be expected to have increased signal during stimulation periods, and "baseline" signal otherwise (Fig 1). While consistent BOLD responses were observed across runs/participants/groups, some regions such as the insula, had increased signal during the stimulation period, whereas other regions had strong peaks at the onset and offset of stimulation, such as the anterior cingulate cortex. Regions of the BS receive input signaling from number of sources, both ascending and descending, and form a complex feed-back network to regulate pain responses depending on a person's situation [1]. This results in complicated BOLD response patterns as the total input signaling (and thus metabolic demand) in each region varies throughout the stimulation paradigm. Nonetheless, regions such as the SC dorsal horn, NRM, NGc, etc. were identified by connectivity analyses. Although the BOLD responses in these regions are complex, the relationships between signal variations and periods of the stimulation paradigm can be seen in retrospect.

Consistent with prior studies [12, 32, 33], this comparison of methods also demonstrated important contributions from individual differences in pain responses. SEM connectivity weighting factors between a number of regions were observed to be significantly correlated with pain ratings (Figs 3 and 4). This demonstrates that results from individual participants are sufficiently accurate for this correlation to be detected. Given that pain responses can vary with emotional and cognitive factors, the ability to identify changes in neural signaling in relation to differences between participants, and even between repeated fMRI acquisitions in the same participant, is important for pain research. The BOLD time-series responses shown in S2 Fig further demonstrate this point, with examples of differences in BOLD responses between participants with higher-, or lower-than-average, pain ratings. The ability to detect variations in BOLD responses which are related to behavioral measures further demonstrates the sensitivity of the data-driven analysis methods that were used for this comparison. This comparison of methods included data from females only, with average ages of 39 years and 36 years for the two study groups, and we did not investigate the effects of sex-differences on pain, or the effects of age, which could introduce even greater variability across participants.

Lastly, an important feature demonstrated by the results of this study is the consistency of responses in regions common to data sets spanning the brain and BS/SC. Results in the thalamus, hypothalamus, and PAG, were obtained from all of the data sets, and are shown in the middle row of S2 Fig. Although the study conditions are not identical, similarities can be observed between the BOLD responses detected with $T_2^*$-weighted GE-EPI in brain fMRI studies, and with $T_2$-weighted HASTE in BS/SC fMRI studies, and in the connectivity values listed in S2–S5 and S7–S9 Tables. Moreover, similar dynamic variations in PAG→Hypothalamus connectivity values are demonstrated in Fig 7. The results obtained with the two acquisition methods are thus equivalent in terms of the conclusions that can be drawn about the regions involved, characteristics of the BOLD responses, and the coordination of signaling between regions.

The results of this comparison show that the model-driven fMRI analysis approach of predicting a BOLD response, and detecting the voxels/clusters with a significant component matching this prediction, has limited effectiveness for pain research studies. Data-driven methods based on connectivity analyses are shown to explain more of the observed signal variations in pain-related regions. These methods are also shown to be better suited to pain research because they allow for dynamic analyses, and accommodate unanticipated variations in responses across conditions or participants. This conclusion is supported by consistent findings across duplicate studies, and across different data acquisition methods. FMRI data acquired with methods optimized for the brain ($T_2^*$-weighted GE-EPI) and for the BS/SC ($T_2$-weighted HASTE) are shown to provide equivalent results. The conclusions that can be drawn from these data sets in terms of the regions involved, the neural signaling between regions, and how signaling varies over the course of the stimulation paradigm, are shown to be consistent. The results indicate that the most sensitive method of fMRI data analysis for pain research may be connectivity based on temporal correlation between time-series responses. However, SEM analysis results provide more information about networks of coordinated regions. Correlation-based and SEM-based connectivity analysis methods are suited to testing different hypotheses, with the former demonstrating which anatomical regions are likely to be involved, and the latter demonstrating the nature of coordinated signaling within a network of predetermined anatomical regions.

The temporal patterns of BOLD responses that were observed with this analysis demonstrate why model responses based on the stimulation paradigm are ineffective for pain research. The observed BOLD responses demonstrate a number of consistent features that are not modeled, including transient responses at the onset and offset of stimulation, which may reflect salience or novelty, and variations leading up to the stimulation period, which may reflect anticipation. Data-driven approaches were able to identify coordinated regions, and the BOLD responses within these regions could then be investigated to show these different aspects of pain responses, and the results also demonstrated individual differences and how responses varied in relation to pain ratings. Data-driven analysis methods are thus better adapted to the variability of neural responses involved with pain processing, and are thus more effective.

## Supporting information

**S1 Fig. Anatomical networks used for structural equation modeling (SEM) in the brain (left) and brainstem and spinal cord (right).** Abbreviations: AC: anterior cingulate cortex; Accum: nucleus accumbens; Amyg: amygdala; C6RD: right dorsal region of the 6$^{th}$ cervical spinal cord segment; DRt: dorsal reticular nucleus of the medulla; Forb: frontal orbital region; HG: heschel's gyrus; Hipp: hippocampus; Hyp: hypothalamus; IC: insular cortex; LC: locus

coeruleus; NGc: nucleus gigantocellularis; NRM: nucleus raphe magnus; NTS: nucleus tractus solitarius; PAG: periaqueductal gray region; PBN: parabrachial nucleus; PC: posterior cingulate cortex; Thal: thalamus.
(TIF)

**S2 Fig. Details of BOLD responses in regions identified by means of the analysis methods, compared across the two sets of study data, for brain and brainstem regions.** BOLD responses shown are averaged across all runs/participants in each study group, except where shown for pain ratings separated between above and below average values. The plots in each frame are offset vertically for clarity. The vertical axis indicates the relative scale for each frame. The horizontal axis indicates time, and plots are shown with the onset of stimulation aligned for each study. The vertical line indicates when participants were informed of which type of stimulation to expect in Study 2. The fit model paradigm to each time-series response is also shown in black on each plot.
(TIF)

**S1 Table. Significant BOLD responses detected by fitting a predicted response to data from brain regions in Studies 1 and 2, by means of a general linear model.** Abbreviations are listed in the caption for S1 Fig.
(DOCX)

**S2 Table. Significant connectivity between regions, based on temporal correlations between BOLD responses, for data from brain regions in Studies 1 and 2.** The values shown are for the epoch spanning the stimulation period. Abbreviations are listed in the caption for S1 Fig.
(DOCX)

**S3 Table. Significant connectivity between regions, based on structural equation modeling, with 1 source for each target (i.e. linear fit between regions), for data from brain regions in Studies 1 and 2.** The values shown are for the epoch spanning the stimulation period. Abbreviations are listed in the caption for S1 Fig.
(DOCX)

**S4 Table. Significant connectivity between regions, based on structural equation modeling, with 2 source regions for each target, for data from brain regions in Studies 1 and 2.** The values shown are for the epoch spanning the stimulation period. Abbreviations are listed in the caption for S1 Fig.
(DOCX)

**S5 Table. Significant connectivity between regions, based on structural equation modeling, with 2 source regions for each target, where at least one connection has connectivity weighting factors that are significantly correlated with pain ratings across participants.** Data are from brain regions in Study 1. Values are listed for the epoch spanning the stimulation period. Abbreviations are listed in the caption for S1 Fig.
(DOCX)

**S6 Table. Significant BOLD responses detected by fitting a predicted response to data from brainstem and spinal cord regions in Studies 1 and 2, by means of a general linear model.** Abbreviations are listed in the caption for S1 Fig.
(DOCX)

**S7 Table. Significant connectivity between regions, based on temporal correlations between BOLD responses, for data from brainstem and spinal cord regions in Studies 1**

**and 2.** Values are listed for the epoch spanning the stimulation period. Abbreviations are listed in the caption for S1 Fig.
(DOCX)

**S8 Table. Significant connectivity between regions, based on structural equation modeling, with 1 source for each target (i.e. linear fit between regions), for data from brainstem and spinal cord regions in Studies 1 and 2.** Values are listed for the epoch spanning the stimulation period. Abbreviations are listed in the caption for S1 Fig.
(DOCX)

**S9 Table. Significant connectivity between regions, based on structural equation modeling, with 2 source regions for each target, for data from brainstem and spinal cord regions in Studies 1 and 2.** Abbreviations are listed in the caption for S1 Fig.
(DOCX)

**S10 Table. Significant connectivity between regions, based on structural equation modeling, with 2 source regions for each target, where at least one connection has connectivity weighting factors that are significantly correlated with pain ratings across participants.** Data are from brainstem and spinal cord regions in Studies 1 and 2. The Z-score is computed from the correlation, R, value, and reflects the significance of the correlation. Only significant values are shown (corrected for multiple comparisons, etc, etc). Values are listed for the epoch spanning the stimulation period. Abbreviations are listed in the caption for S1 Fig.
(DOCX)

## Acknowledgments

We are grateful to Don Brien and Janet Mirtle-Stroman for help with fMRI data collection. All authors confirm that they have no conflicts of interest to disclose with regards to the work presented.

## Author Contributions

**Conceptualization:** Patrick W. Stroman.

**Data curation:** Patrick W. Stroman, Howard J. M. Warren, Gabriela Ioachim, Jocelyn M. Powers, Kaitlin McNeil.

**Formal analysis:** Patrick W. Stroman.

**Funding acquisition:** Patrick W. Stroman.

**Investigation:** Patrick W. Stroman.

**Methodology:** Patrick W. Stroman.

**Project administration:** Patrick W. Stroman.

**Resources:** Patrick W. Stroman.

**Software:** Patrick W. Stroman.

**Supervision:** Patrick W. Stroman.

**Validation:** Patrick W. Stroman.

**Visualization:** Patrick W. Stroman.

**Writing – original draft:** Patrick W. Stroman, Howard J. M. Warren.

**Writing – review & editing:** Patrick W. Stroman, Howard J. M. Warren, Gabriela Ioachim, Jocelyn M. Powers, Kaitlin McNeil.

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
