## [Decision Letter · Decision Letter 0]

20 Oct 2020

PONE-D-20-23531

A Comparison of the Effectiveness of Functional MRI Analysis Methods for Pain Research: The New Normal

PLOS ONE

Dear Dr. Stroman,

Thank you for submitting your manuscript to PLOS ONE. After careful consideration, we feel that it has merit but does not fully meet PLOS ONE’s publication criteria as it currently stands. Therefore, we invite you to submit a revised version of the manuscript that addresses the points raised during the review process.

There are a number of points to address, however, most are relatively minor. In addition to the points raised by the Reviewers, please also state in the manuscript itself that informed consent was obtained from the participants in both studies.

We look forward to receiving your revised manuscript.

Kind regards,

Niels Bergsland

Academic Editor

PLOS ONE

Journal Requirements:

3. Please clarify in your Ethics statement if the authors of this study had access to identifying information. Please also add this information, along with the rest of the Ethics statement, to your Methods section.

Reviewers' comments:

Reviewer's Responses to Questions

**Comments to the Author**

1. Is the manuscript technically sound, and do the data support the conclusions?

Reviewer #1: Yes

Reviewer #2: Yes

2. Has the statistical analysis been performed appropriately and rigorously? 

Reviewer #1: Yes

Reviewer #2: Yes

3. Have the authors made all data underlying the findings in their manuscript fully available?

Reviewer #1: Yes

Reviewer #2: Yes

4. Is the manuscript presented in an intelligible fashion and written in standard English?

Reviewer #1: Yes

Reviewer #2: Yes

5. Review Comments to the Author

Reviewer #1: Stroman et al. present a manuscript of the impact of different fMRI analyses on pain research, describing model-driven and data-driven methods and respective findings. This is a very interesting manuscript that shows how analyses in the neuroimaging field can be useful to investigate mechanisms related to pain. The manuscript is well written and the discussion satisfactorily show the most effective method for fMRI studies of pain. However, I still have some suggestions to improve the manuscript. I remain enthusiastic about the goal of this study and the discussion made by the authors.

1. Methods – Brain fMRI data: The authors stated: “The data were converted to NIfTI format, motion and slice time corrected, co-registered to their anatomical images, and normalized to the MNI template (Montreal Neurological Institute, Montreal, Quebec).” Please describe the preprocessing steps in more detail, including quality control methods to account for motion.

2. Methods – Study 2: Please mention the body location of Study 2’s stimulation.

3. Methods: Please describe the methods and software used to create figures 2, 3 and 4.

4. Discussion: Even though, the study had a consistent sample, i.e. only women, the age differences were significant. Please add some sentences to the discussion pointing out the potential effects of age and also sex on these findings. There is a vast literature discussing sex and age differences in brain regions involved in pain processing.

Reviewer #2: PONE-D-20-23531

Thank you for providing me the opportunity to review the manuscript by Stroman et al comparing the effectiveness of fMRI analysis in the context of brain research. Firstly, I would like to congratulate the authors on a very important and well-designed study, well-written manuscript. Please find below some comments:

• The introduction is lengthy and thus, the authors are encouraged to shorten it. Parts of the introduction can be moved to the discussion.

• Page 3, line 75: Spinal cord and brainstem can be abbreviated as the authors have introduced these abbreviations earlier in the introduction.

• Is there any overlap in the study participants of study 1 and 2? I.e., Are all participants unique or did some take part in both studies?

• What was the rationale to only include female subjects?

• Can the authors provide a table with the individual data for both cohorts? Please add the information regarding assess anxiety, depression, pain catastrophizing, social desirability, and health-related factors. Even if it was not used for the analysis. This table can go to the supplementary material if needed.

• Did the authors assess handedness in their participants? The authors state, that the thermode was placed on the right hand. Was this the dominant hand for everyone?

• The authors state: ‘Only the data from the TSSP condition are used for the present comparison of methods’. What was the rationale to exclude TSSP-C?

• How did the authors correct for multiple comparisons? What method was applied (e.g., Bonferroni)?

• The authors are encouraged to make the code for analysis openly accessible. This would allow researchers to fully understand the analysis conducted, including the small details.

• Even though details on the study participants are described in previous studies, it might be useful for the reader if there is a summary table about the participants in the supplementary material.

• Figure 5: The authors show the R^2 values of their analysis. However, the bar plot is not an ideal choice to visualize these results as it implies a range. Thus, the authors are encouraged to use a dotplot or similar.

• In the supplementary tables, please specify the abbreviations (even though they are explained in the main text) as these are ‘standalone’ objects that should be understandable without the main text.

• The authors show the results from two cross-sectional studies. What is the authors experience in the reliability of such measurments within subjects at different timepoints? Please comment on that.

• Also, this study focuses on healthy controls that undergo experimental pain conditions. What is the authors opinion on the generalizability of this data-driven ad hypothesis driven methods, when applied to clinical pain populations? Also in the context of pain population undergoing experimental pain paradigms.

• Please add a link to the figshare repository, where the data is stored and publicly available, in the manuscript.

6. PLOS authors have the option to publish the peer review history of their article (what does this mean?). If published, this will include your full peer review and any attached files.

Reviewer #1: No

Reviewer #2: No

---

## [Author Response · Author response to Decision Letter 0]

3 Nov 2020

Please see attached document with our detailed responses to each of the editor comments, and points raised by the reviewers.

---

## [Decision Letter · Decision Letter 1]

26 Nov 2020

A Comparison of the Effectiveness of Functional MRI Analysis Methods for Pain Research: The New Normal

PONE-D-20-23531R1

Dear Dr. Stroman,

We’re pleased to inform you that your manuscript has been judged scientifically suitable for publication and will be formally accepted for publication once it meets all outstanding technical requirements.

Kind regards,

Niels Bergsland

Academic Editor

PLOS ONE

Additional Editor Comments (optional):

Reviewers' comments:

Reviewer's Responses to Questions

**Comments to the Author**

1. If the authors have adequately addressed your comments raised in a previous round of review and you feel that this manuscript is now acceptable for publication, you may indicate that here to bypass the “Comments to the Author” section, enter your conflict of interest statement in the “Confidential to Editor” section, and submit your "Accept" recommendation.

Reviewer #1: All comments have been addressed

Reviewer #2: All comments have been addressed

2. Is the manuscript technically sound, and do the data support the conclusions?

Reviewer #1: Yes

Reviewer #2: Yes

3. Has the statistical analysis been performed appropriately and rigorously? 

Reviewer #1: Yes

Reviewer #2: Yes

4. Have the authors made all data underlying the findings in their manuscript fully available?

Reviewer #1: Yes

Reviewer #2: Yes

5. Is the manuscript presented in an intelligible fashion and written in standard English?

Reviewer #1: Yes

Reviewer #2: Yes

6. Review Comments to the Author

Reviewer #1: Authors satisfactorily addressed the reviewers' concerns and the manuscript is ready for publication. I suggest that the manuscript is accepted for publication.

Reviewer #2: The authors have addressed this reviewer's comments appropriately. The manuscript is written well. Thank you.

7. PLOS authors have the option to publish the peer review history of their article (what does this mean?). If published, this will include your full peer review and any attached files.

Reviewer #1: No

Reviewer #2: No

---

## [Editor Report · Acceptance letter]

4 Dec 2020

PONE-D-20-23531R1 

A Comparison of the Effectiveness of Functional MRI Analysis Methods for Pain Research: The New Normal 

Dear Dr. Stroman:

I'm pleased to inform you that your manuscript has been deemed suitable for publication in PLOS ONE. Congratulations! Your manuscript is now with our production department. 

Kind regards, 

on behalf of

Dr. Niels Bergsland 

Academic Editor

PLOS ONE